# Evaluation of Propiophenone, 4-Methylacetophenone and 2′,4′-Dimethylacetophenone as Phytotoxic Compounds of Labdanum Oil from *Cistus ladanifer* L.

**DOI:** 10.3390/plants12051187

**Published:** 2023-03-06

**Authors:** María Espinosa-Colín, Irene Hernandez-Caballero, Celia Infante, Irene Gago, Javier García-Muñoz, Teresa Sosa

**Affiliations:** 1Department of Plant Biology, Ecology and Earth Sciences, Faculty of Science, University of Extremadura, 06006 Badajoz, Spain; 2Department of Anatomy, Cellular Biology and Zoology, Faculty of Science, University of Extremadura, 06006 Badajoz, Spain; 3Department of Forest Resources Technology, Center for Scientific and Technological Research of Extremadura (CICYTEX), 06187 Badajoz, Spain; 4Toxicology Unit, Faculty of Veterinary Medicine, University of Extremadura, 06006 Badajoz, Spain

**Keywords:** phytotoxicity, phenolic compounds, allelopathy, *Cistus ladanifer*, bioherbicides

## Abstract

This is the first study to evaluate the phytotoxic activity of three phenolic compounds present in the essential oil of the labdanum of *Cistus ladanifer*, an allelopathic species of the Mediterranean ecosystem. Propiophenone, 4′-methylacetophenone, and 2′,4′-dimethylacetophenone slightly inhibit total germination and radicle growth of *Lactuca sativa*, and they strongly delay germination and reduce hypocotyl size. On the other hand, the inhibition effect of these compounds on *Allium cepa* was stronger on total germination than on germination rate, and radicle length compared to hypocotyl size. The position and number of methyl groups will affect the efficacy of the derivative. 2′,4′-dimethylacetophenone was the most phytotoxic compound. The activity of the compounds depended on their concentration and presented hormetic effects. In *L. sativa*, on paper, propiophenone presented greater inhibition of hypocotyl size at greater concentrations, with IC_50_ = 0.1 mM, whereas 4′-methylacetophenone obtained IC_50_ = 0.4 mM for germination rate. When the mixture of the three compounds was applied, in *L. sativa*, on paper, the inhibition effect on total germination and the germination rate was significantly greater compared to the effect of the compounds when they were applied separately; moreover, the mixture inhibited radicle growth, whereas propiophenone and 4′-methylacetophenone did not exert such effect when applied separately. The activity of the pure compounds and that of the mixture also changed based on the substrate used. When the trial was conducted in soil, the separate compounds delayed the germination of the *A. cepa* to a greater extent compared to the trial on paper, although they stimulated seedling growth. In soil, *L. sativa* against 4′-methylacetophenone also showed the opposite effect at low concentrations (0.1 mM), with stimulation of germination rate, whereas propiophenone and 4′-methylacetophenone presented a slightly increased effect.

## 1. Introduction

In recent years, thanks to different technological innovations and advances, agricultural production has been improved, expanding the irrigated area and thus obtaining greater profit; however, this has also brought major problems related to the deterioration of the natural resources, such as deforestation, soil degradation, biodiversity loss, desertification and contamination due to the use of artificial chemical products [1].

The continuous exponential growth of the human population implies the further exploration of new advances that continue to increase agricultural production. Therefore, it is essential to use pesticides; the question is: where can we find safe products that do not compromise the environment and health? One of the emerging strategies to attain a more sustainable future is to search for solutions in nature itself. Nature has already solved all the challenges we face today; its failures have become fossils, and what surrounds us in the present time is the key to survival. Mimicking these measures is known as biomimicry [2].

In ecosystems where resources are scarce, some plants synthesise phytotoxic compounds, with which they manage to inhibit the growth of other species around them, thereby preventing future competition for resources. This phenomenon is called allelopathy, and it has been widely studied [3]. In addition, other studies have also shown the phytotoxic effect of the essential oils of these allelopathic species on different weeds. This negative effect of essential oils on germination is related to the presence of allelochemical compounds [4,5,6]. Currently, allelopathic compounds may constitute a natural source of bioherbicides [7], but often these molecules are complex and difficult to reproduce. To avoid this problem, we must focus the search on low molecular weight structures such as phenolic compounds. Although the phytotoxic effect of phenolic compounds is well known [8,9,10], many of them have not been evaluated to date, due to their diversity and the difficulty in isolating or synthesising them. Acetophenone, due to its chemical, biological, pharmacological, and allelochemical properties, can be a good candidate in different fields such as agronomy. Acetophenone is a common secondary metabolite in plant species [11]. Different studies show that its derivatives are involved in defense mechanisms [11]. Some studies on allelopathic plants have found that acetophenone derivatives such as xanthoxylin and acetosyringone can be good herbicides [12,13,14]. Another example of an allelopathic species where acetophenone is present is *Cistus ladanifer* Linneo (1753). This shrub is a typical species of the Mediterranean ecosystem that can eventually form monospecific communities known as rock rose thickets. When this species is exploited for the extraction of essential oil and/or labdanum, plant richness increases immediately in the area, although, due to the seed bank preserved in the soil and the regenerative capacity of this species, such plant richness decreases progressively after two years [15]. A large number of studies show that this reduction of biodiversity in rock rose thickets may be due to the allelopathic potential of this species [16,17,18,19]. In addition, its essential oil also presents phytotoxic activity [20,21]. The labdanum of its leaves and its essential oil contain a concentrated amount of secondary metabolites with different properties [22]. An important group of this labdanum is constituted of phenolic compounds [21,23,24,25,26,27,28]. Acetophenone is also found in the essential oil of labdanum. Three derivatives of acetophenone (propiophenone, 4′-methylacetophenone, and 2′,4′-dimethylacetophenone) have been identified in the neutral part of labdanum oil [29,30]. Labdanum oil contains more than 300 constituents, with 186 compounds (154 from the neutral part and 32 from the acidic part) representing about 95% of this oil [30,31].

Propiophenone (Figure 1A), 4′-methylacetophenone (Figure 1B), and 2′,4′-dimethylacetophenone (Figure 1C) are simple and very similar phenolic compounds. All three of them present an acetophenone backbone, with a methyl group in different positions. Phenolic compounds present variable phytotoxicity depending on their formula, and small differences may result in different phytotoxic activity [9]. The inhibitory action of a phenolic compound may also change based on the concentration applied [19,22]. In some cases, even hormetic effects can be observed [32,33,34,35]. Furthermore, in soil, phenolic compounds can undergo processes of transportation, retention, and transformation, which can also influence the final result of bioactivity [36]. Moreover, it is worth highlighting that the action of allelopathic plants is commonly caused by the mixture of several allelochemicals that are found in the soil at very low concentrations [9].

Nowadays, thanks to their commercial availability, propiophenone, 4′-methylacetophenone, and 2′,4′-dimethylacetophenone can be easily evaluated through a standardised phytotoxicity bioassay. To date, no adverse effects have been reported in humans from these three compounds, and 4′-methylacetophenone is used in disinfectants, pest-control products, perfumes, and cosmetics [37].

A very important aspect that allows the introduction of allelochemicals in the natural management of weeds is the knowledge of target plants and the exact chemicals responsible for the interaction [38]. Therefore, the aim of this study was to evaluate the phytotoxic activity of propiophenone, 4′-methylacetophenone and 2′,4′-dimethylacetophenone through a standard bioassay that contemplates all the possible variables. To this end, seeds of *Lactuca sativa* L., as a representative of monocotyledons, and *Allium cepa* L., as a representative of dicotyledons, were used, due to their sensitivity. In addition, their rapid, complete, and uniform germination grants reliability and duplicity to the assay [39,40,41]. These seeds were planted on paper and in soil, and they were watered with different concentrations of each of the compounds separately and a mixture of all three. Their phytotoxic activity was assessed by measuring their effect on germination and seedling growth [42]. The results may contribute to finding possible structures that serve as sources of natural bioherbicides that are safer and less contaminant.

## 2. Results

### 2.1. Effect of Propiophenone, 4′-Methylacetophenone and 2′,4′-Dimethylacetophenone on the Germination of Lactuca sativa L.

The results showed that propiophenone, 4′-methylacetophenone, 2′,4′-dimethylacetophenone and the mixture of the three compounds behaved in a similar manner, the three compounds separately, and their mixture inhibited germination rate (%GR) to a greater extent than total germination (%GT), and, lastly, the greater the concentration, the greater the inhibition effect (Figure 2). Both in soil and on paper, the three compounds separately and their mixture at 1 mM slightly inhibited total germination, except when the mixture was applied on paper, which showed full inhibition of total germination. The results for germination rate showed that propiophenone, 2′,4′-dimethylacetophenone and the mixture strongly inhibited %GR on paper and in soil, whereas 4′-methylacetophenone produced a lower effect; moreover, when the soil was watered with 0.1 mM of 4′-methylacetophenone, the opposite effect was obtained, showing significant stimulation of this parameter.

At the three analysed concentrations, propiophenone strongly inhibited germination rate (over 80%) both on paper and in soil, and it also significantly inhibited total germination (over 10%) in soil.

On paper and in soil, 4′-methylacetophenone significantly inhibited total germination at 0.5 mM and 1 mM. On paper, this compound also significantly inhibited germination rate with all analysed concentrations, showing a significant positive correlation (R^2^ = 0.99), i.e., the greater the concentration, the greater the inhibition. The effective concentration required to induce half-maximal inhibition (IC_50_) was 0.4 mM. It is worth highlighting that, when the soil was watered with 1 mM of 4′-methylacetophenone, the germination rate was significantly inhibited, whereas the concentration of 0.1 mM stimulated this parameter.

In the assays with 2′,4′-dimethylacetophenone at 1 mM, %GT was significantly inhibited both on paper and in soil. This inhibition was only 7.5% and 4.5% on paper and in soil, respectively; however, as in the case of propiophenone, the germination rate was strongly inhibited at all three concentrations analysed, both on paper and in soil, showing no significant differences between the three concentrations.

The mixture inhibited total germination and germination rate at all analysed concentrations both on paper and in soil. When the mixture of the three compounds was applied on paper at 1 mM, %GT, and %GR were inhibited to a significantly greater extent compared to the inhibition obtained with the compounds separately. However, in soil, the inhibition of %GT produced by the mixture at 1 mM was only significantly greater than that obtained with 2′,4′-dimethylacetophenone separately, whereas %GR was inhibited to a significantly greater extent with the mixture than with 4′-methylacetophenone and 2′,4′-dimethylacetophenone separately.

### 2.2. Effect of Propiophenone, 4′-Methylacetophenone and 2′,4′-Dimethylacetophenone on the Seedling Growth of Lactuca sativa L.

The results for radicle and hypocotyl growth indicate that propiophenone, 4′-methylacetophenone, 2′,4′-dimethylacetophenone, and the mixture played a similar role on paper, the three compounds separately and the mixture inhibited hypocotyl length (%Hypocotyl length) to a greater extent than radicle length (%Radicle length), and greater inhibition was obtained at greater concentrations (Figure 3). Both in soil and on paper, the three compounds separately and their mixture slightly inhibited radicle growth, except for the soil assay with 2′,4′-dimethylacetophenone, which produced an inhibition of over 42%. The three compounds separately and their mixture strongly inhibited hypocotyl growth on paper; on the other hand, in soil, although the effect was greater with 2′,4′-dimethylacetophenone, it decreased with propiophenone and the mixture, obtaining the opposite effect when 4′-methylacetophenone was applied, with significant stimulation of this index at all three concentrations analysed.

When the assay was carried out in soil, propiophenone significantly inhibited radicle and hypocotyl growth at the three analysed concentrations. When the assay was performed on paper, radicle length was not significantly affected, whereas hypocotyl size decreased by over 52%. This inhibition was greater at higher concentrations, being significantly dependent on concentration (R^2^ = 0.98) and obtaining an IC_50_ of 0.1 mM.

Both on paper and in soil, 4′-methylacetophenone significantly inhibited radicle length at 1 mM, but not at lower concentrations. When the assay was conducted in paper, hypocotyl size was strongly inhibited at all three concentrations. This inhibition showed a positive correlation with concentration (R^2^ = 0.99), obtaining an IC_50_ lower than 0.1 mM. Nevertheless, when the assay was carried out in soil, hypocotyl growth was stimulated at all three concentrations.

At all concentrations, both on paper and in soil, 2′,4′-dimethylacetophenone significantly inhibited radicle and hypocotyl growth, observing a greater effect for the assays in soil.

The mixture of propiophenone, 4′-methylacetophenone and 2′,4′-dimethylacetophenone inhibited radicle and hypocotyl length at all three concentrations on paper and in soil, except when the paper was watered with 0.1 mM, where radicle length was significantly greater than in the control. On paper, at 1 mM, radicle and hypocotyl length % was cero, since, at that concentration of the mixture, the seeds did not germinate; at 0.5 mM, the mixture inhibited radicle growth to the same extent as 2′,4′-dimethylacetophenone, whereas propiophenone and 4′-methylacetophenone separately did not show such inhibition; at 0.1 mM, the mixture stimulated radicle growth to the same extent as propiophenone alone. On paper, the mixture inhibited %Hypocotyl length to a lesser extent than the isolated compounds. In soil, the inhibitory effect of the mixture on radicle and hypocotyl length was significantly greater than that of the compounds separately, except for 2′,4′-dimethylacetophenone.

### 2.3. Effect of Propiophenone, 4′-Methylacetophenone and 2′,4′-Dimethylacetophenone on the Germination of Allium cepa L.

When the assay was carried out with *Allium cepa*, it was also observed that the inhibition effects of the separate compounds were greater at higher concentrations, although the behaviour was slightly different (Figure 4). Unlike in the case of *Lactuca sativa*, propiophenone, 4′-methylacetophenone, and 2′,4′-dimethylacetophenone inhibited total germination (%GT) to a greater extent than germination rate (%GR). In this case, both in soil and on paper, the three compounds separately at 1 mM strongly inhibited total germination and germination rate, and, in contrast to the results obtained with *Lactuca sativa*, 4′-methylacetophenone did not present hormetic effects, but propiophenone did.

At all three analysed concentrations, propiophenone inhibited %GT more strongly than %GR in soil with respect to the paper assay. Moreover, when the paper was watered with a concentration of 1 mM, the germination rate was significantly inhibited, whereas, at 0.1 mM, it was stimulated by over 20%.

In both paper and soil, 4′-methylacetophenone significantly inhibited total germination and germination rate at all three analysed concentrations. At 1 mM, the inhibition was very strong: over 90% on paper and over 45% in soil. Furthermore, in soil, this inhibition depended on the concentration, presenting a positive correlation (R^2^ = 0.99 for %GT, R^2^ = 0.98 for %GR). The effective concentration required to induce the half-maximal inhibition of %GT and %GR (IC_50_) was 0.1 mM. 

When the assay was performed with 2′,4′-dimethylacetophenone, a similar behaviour was observed with respect to propiophenone: the inhibition effects of %GT and %GR are greater in soil than on paper, and %GR stimulation was observed when the paper was watered with 0.1 mM. In soil, 2′,4′-dimethylacetophenone strongly inhibited %GT and %GR at the three analysed concentrations. This effect depended on the concentration, presenting a significant positive correlation (R^2^ = 0.98 for %GT, R^2^ = 0.99 for %GR). The effective concentration required to induce the half-maximal inhibition of %GT (IC_50_) was 1 mM, and 0.1 mM for %GR.

On its part, the mixture of compounds also presented a different behaviour with respect to that observed in *Lactuca sativa*. On paper, the mixture inhibited total germination and germination rate at the highest concentration; however, in soil, the effect was the opposite. In soil, at the highest and lowest concentrations, %GT and %GR were significantly stimulated by over 26%. When the mixture of the three compounds was applied on paper at 0.5 mM, %GR was inhibited, which was not observed with the separate compounds, and, at 0.1 mM, %GR was inhibited significantly more strongly with the mixture than with propiophenone and 2′,4′-dimethylacetophenone separately.

### 2.4. Effect of Propiophenone, 4′-Methylacetophenone and 2′,4′-Dimethylacetophenone on the Seedling Growth of Allium cepa L.

The effect on the seedling growth of *Allium cepa*, when watered with propiophenone, 4′-methylacetophenone, 2′,4′-dimethylacetophenone or the mixture, was different from that observed in *Lactuca sativa*. On paper, the three separate compounds and the mixture inhibited radicle length more strongly (%Radicle length) than hypocotyl length (%Cotyledon length), being greater at higher concentrations (Figure 5). However, in soil, the tendency observed was the opposite. The three compounds separately and the mixture stimulated seedling growth at the three analysed concentrations; only at 0.1 mM of propiophenone and 1 mM of 4′-methylacetophenone, hypocotyl length was significantly inhibited.

On paper, radicle and hypocotyl growth were strongly inhibited when watered with concentrations above 0.5 mM, although it was significantly stimulated when watered with a concentration of 0.1 mM. In soil, radicle size was not significantly affected, unlike hypocotyl size, which decreased by over 25% at 0.1 mM and increased by over 20% at higher concentrations.

The same was observed with 4′-methylacetophenone. When the assays were carried out on paper, radicle, and cotyledon length were significantly inhibited at the three analysed concentrations, whereas the opposite effect was observed in soil.

On paper, 2′,4′-dimethylacetophenone significantly inhibited radicle growth at 1 mM, although at 0.1 mM it significantly stimulated hypocotyl length. In soil, this compound stimulated the seedling growth of *Allium cepa* at all concentrations analysed.

On paper, the mixture of propiophenone, 4′-methylacetophenone and 2′,4′-dimethylacetophenone strongly inhibited radicle and hypocotyl length at all three concentrations; however, at 0.1 mM, hypocotyl length was slightly greater than in the control. The effect of the mixture on radicle and hypocotyl length was significantly greater than that of propiophenone or 2′,4′-dimethylacetophenone separately. On the other hand, in soil, the mixture of compounds significantly stimulated seedling growth at the three concentrations analysed, with this stimulation being greater than that observed when each of the compounds were applied separately.

## 3. Discussion

The appearance of resistance in weeds and the contamination of natural areas are currently the main problems derived from the massive use of herbicides. To prevent health problems and a decrease in the productivity of crops, it is necessary to find safer and more innovative products with new target sites [43,44]. This study proposes searching for phytotoxic compounds in essential oils of allelopathic species, such as *Cistus ladanifer*. The process of natural selection has given rise to allelochemicals with great structural diversity and adapted to interact with specific target sites. Therefore, through the study of these natural products, it may be possible to develop new tools for the development of new herbicides. However, in many cases, these compounds are complex and difficult to synthesise at an industrial scale. Distillation is the most common method for the extraction of plant essential oils. This method is a process of physical separation, not a chemical reaction. It consists in heating the plant until its most volatile components reach the vapour phase; then, the vapour is cooled down until these components are recovered in liquid form through a condensation process. In this process, complex molecules are hydrolysed into simpler molecules. This may pose a source of simpler derivatives of allelochemicals. If these derivatives preserve their reaction site, their activity may be the same or even greater. Evaluating the phytotoxic activity of simple and low-molecular-weight molecules may help to find a structure that can be used as a backbone for new bioherbicides. An important group of natural phytotoxic compounds is constituted of phenolic compounds [10,12,45]. In the essential oil of *Cistus ladanifer* labdanum, this group of compounds is very abundant and diverse in the exudate of its leaves and photosynthetic stems, although the impossibility of isolating some of them in sufficient amounts has prevented their evaluation. Currently, the commercial availability of three phenolic compounds present in this labdanum oil of *C. ladanifer*, i.e., propiophenone, 4′-methylacetophenone, and 2′,4′-dimethylacetophenone, allows measuring their phytotoxic activity through a germination bioassay in Petri dishes. The results of the present study showed that propiophenone, 4′-methylacetophenone, and 2′,4′-dimethylacetophenone may act as regulators of germination and growth in *Lactuca sativa* L. and *Allium cepa* L.

Although the assay on paper and in soil showed that 4′-methylacetophenone and 2′,4′-dimethylacetophenone slightly inhibited the germination of *L. sativa* at the highest concentration, on paper, all three analysed compounds strongly delayed germination and hypocotyl length; moreover, 2′,4′-dimethylacetophenone also inhibited radicle growth at concentrations of over 0.5 mM. These effects on the germination rate and seedling growth of *L. sativa* could compromise the survival of the seedling and thus be as harmful as the inhibition of germination [46]. This behaviour is similar to that shown by other phenolic compounds analysed in previous studies [10,12,45], which can be attributed to the multiple effects that this group of compounds may have on the processes of photosynthesis, respiration, and transpiration, as well as on the cell cycle [47,48,49,50]. Another characteristic of phenols is that their inhibitory action depends on concentration [9,35,51]. In the case of *L. sativa*, the concentration of 4′-methylacetophenone was positively correlated with %GR and %Hypocotyl length, showing that, the greater the concentration, the greater the inhibition of germination rate and hypocotyl growth. The inhibitory effect of propiophenone on hypocotyl size was also positively correlated with concentration. Furthermore, it is known that some phenolic compounds act as regulators of plant growth, presenting hormesis; that is, their effects are either stimulating or inhibitory depending on their concentration [35,52,53]. This behaviour was shown by 4′-methylacetophenone on %GR of *L. sativa*, when the assay was conducted in soil, obtaining inhibitory and stimulating effects at 1 mM and 0.1 mM, respectively.

When the assay was performed with seeds of *Allium cepa*, the results were somewhat different. Unlike in the case of *Lactuca sativa*, propiophenone, 4′-methylacetophenone, and 2′,4′-dimethylacetophenone showed stronger inhibition of total germination (%GT) than of germination rate (%GR), and radicle length (%Radicle length) compared to hypocotyl length (%Cotyledon length). In this case, propiophenone and 2′,4′-dimethylacetophenone presented hormetic effects when the trial was conducted on paper. This species-dependent activity has been observed with other phenolic compounds present in *C. ladanifer* [35,51]. Monocotyledons are usually more resistant to allelochemicals than dicotyledons [54]. Both in soil and on paper, the most phytotoxic compound against *Lactuca sativa* was 2′,4′-dimethylacetophenone. In paper, 1 mM of 4′-methylacetophenone presented a strong inhibition of %GT and %GR of *Allium cepa*; in soil, this compound also strongly inhibited these parameters at the three concentrations analysed, although, in soil, 2′,4′-dimethylacetophenone was the compound that presented the greatest phytotoxic effect on *A. cepa*. The analysed compounds, as in the case of any phenolic compound, have a hydroxyl group bound to a benzene molecule, and they only differ from each other in the binding of methyl groups. Their effects are different due to these small structural differences. In the analysed parameters, 2′,4′-dimethylacetophenone produced the greatest inhibition, which suggests that having two methyl groups may increase the activity of the molecule. Studies with 3,4-dihydroxyacetophenone isolated from *Picea schrenkiana* Fisch. et Mey. have shown that this compound significantly inhibits the germination of lettuce seeds at concentrations of 1 mM and 0.5 mM. In this case, the inhibitory effect of this compound was greater on radicle growth than on hypocotyl growth. Similarly, 3,4-dihydroxyacetophenone also exerted hormesis on the growth of rice radicles and radish hypocotyls [55]. Other studies on acaricide activity with compounds derived from 2′-hydroxy-4′-methylacetophenone identified in *Angelica koreana* Maxim. radicles have shown that, in terms of structure-activity relationships, the activity changed after the introduction of hydroxyl and methyl groups in the acetophenone backbone. In a contact toxicity bioassay, 3′-methylacetophenone was more effective as an acaricide against *Dermatophagoides farinae* than 2′-methylacetophenone and 4′-methylacetophenone [56]. Several studies have shown that 4-hydroxy-(2H)-1,4-benzoxazin-3(4H)-one, or DDIBOA, stands out for its phytotoxic action [57,58,59], and it may be used as the basis for the development of new herbicides since it is considered to be the most promising structure for the search of new compounds based on the backbones of benzoxazinones [60]. Its structure also presents a benzene ring, and the studies that have been conducted with some of its derivatives, such as 6-Cl-4-hydroxy-(2H)-1,4-benzoxazin-3(4H)-one and 8-Cl-4-hydroxy-(2H)-1,4-benzoxazine-3(4H)-one, show greater activity and selectivity, respectively, exhibiting greater phytotoxicity than the compounds that do not have substituents in their aromatic ring [61,62]. Other studies show that other derivatives such as o-nitroacetophenone inhibit the germination of *Amaranthus tricolor* L. [63]. The herbicide mechanism of benzoxazinoids is still unclear, although it is known that these compounds interfere with electron transport and ATPase activity in mitochondria, as well as with the functions of H^+^-ATPase in the plasma membrane [64]. On their part, phenolic compounds increase the production of reactive oxygen species and, consequently, produce oxidative stress [65,66]. Moreover, it has been recently known that carbonyl, aromatic compounds, and azaindole are key interactions of the inhibitors of the enzyme 4-hydroxyphenylpyruvate dioxygenase [63].

In most cases, allelopathy by phenolic compounds is probably never due to a single substance. These compounds are not synthesised separately in plants, and they appear as a mixture. Their individual concentrations in the soil are usually very low, and their phytotoxicity seems to depend on the additive effects of each compound in the mixture [33]. Thus, in order to evaluate the possible interactions among the analysed compounds, they were applied as a mixture, and their effects were analysed. The results showed that, when the mixture of the three compounds was applied to *L. sativa* on paper at 1 mM, the inhibition of %GT and %GR was significantly greater than when the compounds were applied separately. At 0.5 mM, the mixture inhibited radicle growth to the same extent as 2′,4′-dimethylacetophenone, whereas propiophenone and 4′-methylacetophenone separately did not show such inhibition; however, the mixture inhibited % Hypocotyl length to a lesser extent than the isolated compounds. When applied to *Alium cepa* on paper, the mixture showed a greater inhibition effect 2′,4′-dimethylacetophenone, and propiophenone separately, whereas 4′-methylacetophenone showed a greater inhibition effect than the mixture. This behaviour is in line with other studies on volatile compounds that showed a considerable increase in phytotoxicity when applied jointly [67].

Another factor to be considered is the substrate on which the bioassay was carried out. In the soil, phenolic compounds can undergo processes of transportation, retention, and transformation, which, in turn, may influence their availability and action [36].

When the bioassay was conducted on a commercial substrate, in the two species analysed, propiophenone, which did not inhibit germination on paper, did inhibit germination in this substrate at all three analysed concentrations. In *Lactuca sativa*, the inhibition effect of 2′,4′-dimethylacetophenone was greater on all the parameters studied when the assay was carried out in soil; however, in *A. cepa*, although this compound in soil also inhibits %GT and %GR to a greater extent, %radicle length and %Cotyledon length are stimulated. On its part, 4′-methylacetophenone also changed its activity when the assay was performed on paper with *L. sativa*, delaying germination and inhibiting hypocotyl growth at the three analysed concentrations; on the other hand, in soil, this compound stimulated these parameters at the three concentrations, except at 1 mM on %GR. When the assay was conducted with *A. cepa*, 4′-methylacetophenone increased the inhibition of %GT and %GR, although the effect on seedling growth was attenuated and even stimulated at low concentrations. The mixture also seemed to change its action in soil, observing, in the case of *L. sativa*, inhibition of %radicle length at 0.1 mM, whereas, on paper, stimulation was obtained, and, in the case of *A. cepa*, inhibition was observed on paper, while stimulation was observed in soil. These activity changes based on the substrate have also been observed in other phenolic compounds [35,51]. Recent studies show that complex synergistic interactions between volatile and phenolic compounds underlie the efficacy of the allelopathic residues aggregated to the soil for weed control [68]. Propiophenone has moderate mobility in the soil, but also high volatilisation, and, in the vapour phase, it is degraded in the atmosphere by reacting with photochemically-produced hydroxyl radicals with an estimated half-life of approximately 5 days [69]. It is also known that acetophenone can be oxidised to phenol by *Aspergillus* species [70]. This possible, simple, and rapid degradation in the soil of the analysed compounds will influence their availability and concentration, which will affect their activity; however, this also implies that the phytotoxic compound is only found in the environment during the period of time that is adequate for its application, reducing or eliminating the risk of contamination. Therefore, phytotoxicity bioassays must be carried out in the substrates to which they will be applied.

Natural products with high phytotoxic activity and new mechanisms of action could be good options to constitute the basis of new herbicides, although, in most cases, their formulae are too complex to be produced, and the simplification of their structures often reduces their activity [71]. Thus, it is interesting to focus on simple, low-molecular-weight compounds of rapid degradation and low accumulation in the soil, in order to prevent their possible influence on non-target organisms. These characteristics are found in simple phenolic compounds [8,9,10]. If we search for these types of allelochemicals in essential oils of allelopathic plants, we will find natural products that, due to natural selection, have already been optimised for this activity. It is necessary to evaluate their activity in order to determine the concentration that must be applied in the soil to obtain the desired effect on the target weeds. It is important to highlight that the phytotoxic effects of the allelochemicals observed on paper do not allow extending the results to the field conditions, due to the simultaneous appearance of biotic and abiotic factors that may influence the activity of the mixtures or pure allelopathic compounds [72,73]. Moreover, it is also important to study the effect of formulations of mixtures that allow finding the way of enhancing their activity and attacking several mechanisms of action simultaneously, with the aim of addressing weed resistance. 

This is the first study to report that propiophenone, 4′-methylacetophenone, and 2′,4′-dimethylacetophenone have phytotoxic activity and may regulate plant growth depending on the concentration applied and the substrate used. These compounds identified in the essential oil of labdanum from *Cistus ladanifer* may provide relatively simple structures, which allows them to be used as bioagents in weed control. The position and number of methyl groups affected the efficacy of the compound. This finding is consistent with other studies considering the molecular design of new acetophenones that are active against weeds [11,63]. Research on the structure-activity relationship of these types of allelochemicals and their derivatives based on the presence of methyl groups and other functional groups could lead to the development of less polluting and more specific bioherbicides. Furthermore, if their structure is simple, their synthesis and production will be easier.

## 4. Materials and Methods

### 4.1. Plant and Substrate Sources

To carry out the bioassay, seeds of *Lactuca sativa* variety romaine verte maraîchère (Vilmorin jardin—CS70110—38291 St Quentin Fallavier Cedex—France) were used as representatives of dicotyledons. This species is ideal for adequately showing the allelochemical effects on the germination processes [40,74]. Moreover, *Lactuca sativa* is recommended by the US EPA (United States Environmental Protection Agency) for phytotoxicity tests, being among the most sensitive species [75]. *Allium cepa* variety Bianca di Maggio (Vilmorin Jardin—CS70110—38291 St Quentin Fallavier Cedex— France) were used as monocotyledons. *Allium cepa* has proved to be the most useful and has repeatedly been suggested as a standard test material in the list of species historically used in plant testing by the International Organization for Standardization [76] and by the Organisation of Economic Cooperation and Development [77]. The total germination of these seeds was over 98%. 

The activity of the allelochemicals can vary according to the planting substrate; for this reason, the experiment was carried out on Whatman No. 118 paper and a commercial substrate. The commercial substrate was of universal type, based on 95% peat, 5% green compost, and 1.3 Kg/m^3^ fertilizer: 12N + 12P + 17K (Geolia, Aki Bricolage España S.L. B-839857—Spain). The commercial universal soil presented the following characteristics: organic matter per dry matter (60%), electrical conductivity (40 mS/m), apparent dry density (320 g/L), grain size (0–20 mm), and pH 5.5–6.5.

None of the materials (seeds and substrate) was sterilised before the experiment.

### 4.2. Phytotoxic Activity Test

Chemically pure reagents (propiophenone, 4′-methylacetophenone and 2′,4′-dimethylacetophenone of >99% purity) were obtained from Sigma-Aldrich (Merck KGaA, Darmstadt, Germany). Different solutions were prepared with Milli-Q water from each component separately. The highest concentration was 1 mM, which is the maximum recommended concentration in allelopathic bioassays [78]. The minimum concentration was 0.1 mM, which was the lowest concentration to show the dissipation of the effect of these compounds. In addition, solutions of 0.5 mM were also prepared. For the mixture of the three phenols, three solutions were prepared with equimolar concentrations of each of the compounds at 1 mM, 0.5 mM, and 0.1 mM. To eliminate the effects of pH, we measured these parameters for each solution. The pH varied between 6 and 6.3 from one solution to another. There were no significant differences in pH among solutions.

The study was carried out in Petri dishes with Whatman paper or 25 g of commercial substrate. Four replicates were conducted for each assay, with 50 seeds of *Lactuca sativa* in each repetition (200 in total). They were watered with 5 mL and 16 mL of the different solutions for the paper and soil plates, respectively. The same amount of Milli-Q water was applied to the control. To prevent evaporation, the plates were sealed with Parafilm and were kept in a germination chamber for 5 days at 22 °C and a photoperiod of 15 h of light and 9 h of darkness.

### 4.3. Measured Indices to Quantify the Phytotoxic Effect 

Germinated seeds were counted daily and total germination (%TG) [35,51] and germination rate (%GR) [69,75] were calculated according to Equations (1) and (2).
(1)$$\%\mathrm{TG}=\frac{N_{T}\cdot 100}{N},$$
where N_T_ is the average number of germinated seeds in each treatment and N is the average number of seeds in the control.
(2)$$\mathrm{GR}=\left(N_{1}\cdot 1\right)+\left(N_{2}-N_{1}\right)\cdot \frac{1}{2}+\left(N_{3}-N_{2}\right)\cdot \frac{1}{3}+\ldots \left(N_{n}-N_{n-1}\right)\cdot \frac{1}{n}\;,$$
where N_1_, N_2_, N_3_, … N_n−1_, N_n_ are the proportions of germinated seeds obtained in the first (1), second (2), third (3), …, (n − 1), (n) days (for lettuce it was n = 5 and for onion n = 6). All results are expressed as a percentage relative to the control (%GR).

On the last day of the experiment, 10 seedlings per Petri dish were randomly selected. Their radicle and hypocotyl length were measured [79] and the median was expressed as a percentage relative to the control, thus: (3)$$\%Radicle\;length=\frac{treatment\;root\;length}{control\;root\;length}\cdot 100$$
(4)$$\%Hypocotyl\;length=\frac{treatment\;cotyledon\;length}{control\;cotyledon\;length}\cdot 100$$

### 4.4. Statistical Analysis

The significance level of the comparisons among treatments was estimated using the Mann-Whitney U test. The differences were considered significant when *p* < 0.05. The interrelationships between germination and seed growth with the concentration of phenolic compounds were determined by Pearson′s determination coefficient. The effective concentrations required to induce half-maximal inhibition of growth (IC_50_) were calculated according to the linear relationship between concentration and percent inhibition or stimulation of plant growth. All statistical analyses were conducted using the statistical software IBM SPSS Statistics version 26.

## Figures and Tables

**Figure 1 plants-12-01187-f001:**
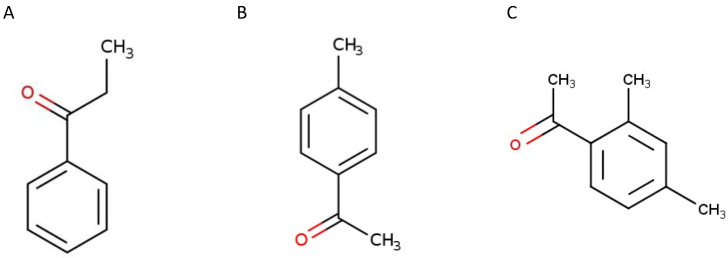
Chemical structure of propiophenone (**A**), 4′-methylacetophenone (**B**), and 2′,4′-dimethylacetophenone (**C**). Source: https://echa.europa.eu/ (accessed on 3 January 2023).

**Figure 2 plants-12-01187-f002:**
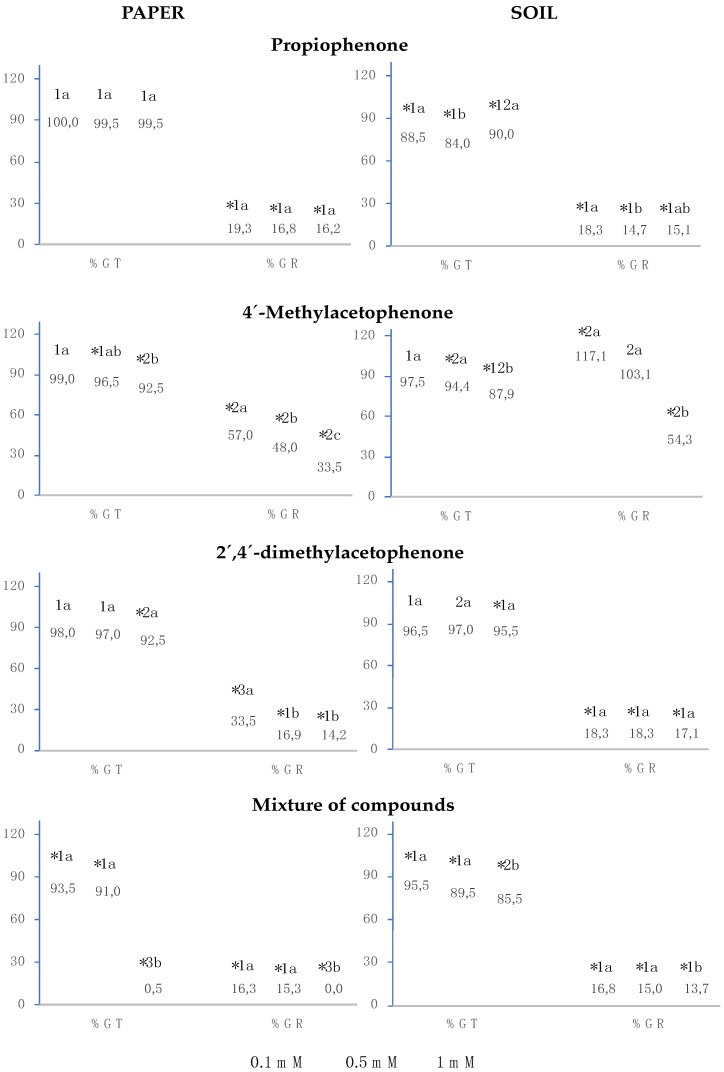
The effect of different concentrations of propiophenone, 4′-methylacetophenone, 2′,4′-dimethylacetophenone, and the mixture of the three compounds on the total germination (%GT) and germination rate (%GR) of *Lactuca sativa*, expressed as the percentage relative to the control. Four replicates of each treatment were performed (n = 4 × 50 = 200 seeds in total for each solution). * Significantly different from the controls; 1, 2, 3: differences numbers indicate significant differences between treatments of the same index and for each concentration. a, b, c: differences in small letters indicate significant differences between concentrations of the same index and for each treatment. *p* < 0.05 (Mann–Whitney U test).

**Figure 3 plants-12-01187-f003:**
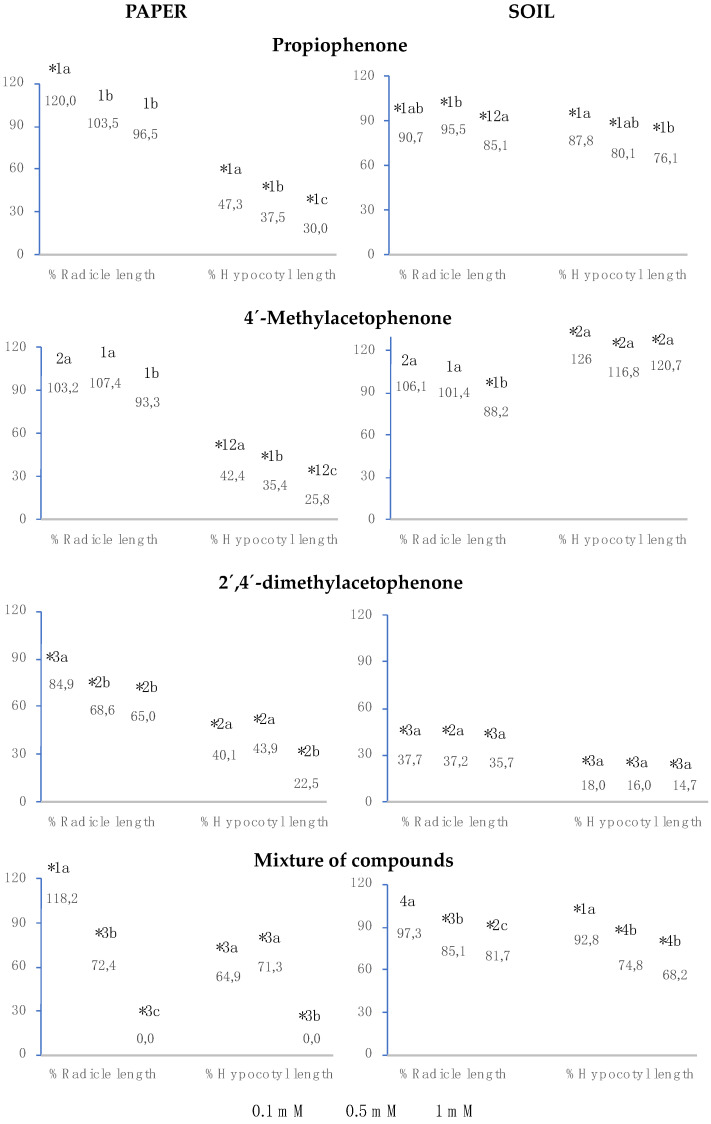
The effect of different concentrations of propiophenone, 4′-methylacetophenone, 2′,4′-dimethylacetophenone, and the mixture of the three compounds on the radicle and hypocotyl length of *Lactuca sativa*, expressed as the percentage relative to the control. Four replicates of each treatment were performed (n = 4 × 50 = 200 seeds in total for each solution). * Significantly different from the controls; 1, 2, 3, and 4: differences numbers indicate significant differences between treatments of the same index and for each concentration. The a, b, c: differences in small letters indicate significant differences between concentrations of the same index and for each treatment. *p* < 0.05 (Mann–Whitney U test).

**Figure 4 plants-12-01187-f004:**
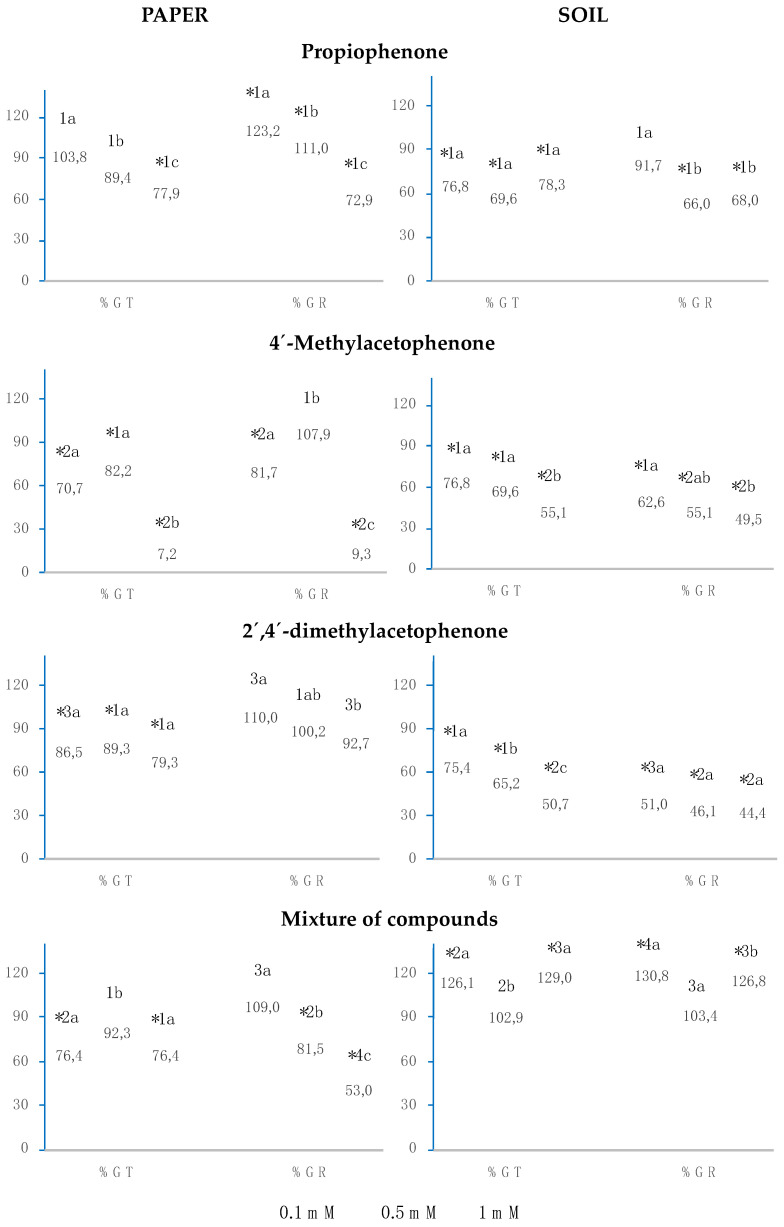
The effect of different concentrations of propiophenone, 4′-methylacetophenone, 2′,4′-dimethylacetophenone, and the mixture of the three compounds on the total germination (%GT) and germination rate (%GR) of *Allium cepa*, expressed as the percentage relative to the control. Four replicates of each treatment were performed (n = 4 × 50 = 200 seeds in total for each solution). * Significantly different from the controls; 1, 2, 3, and 4: differences numbers indicate significant differences between treatments of the same index and for each concentration. The a, b, c: differences in small letters indicate significant differences between concentrations of the same index and for each treatment. *p* < 0.05 (Mann–Whitney U test).

**Figure 5 plants-12-01187-f005:**
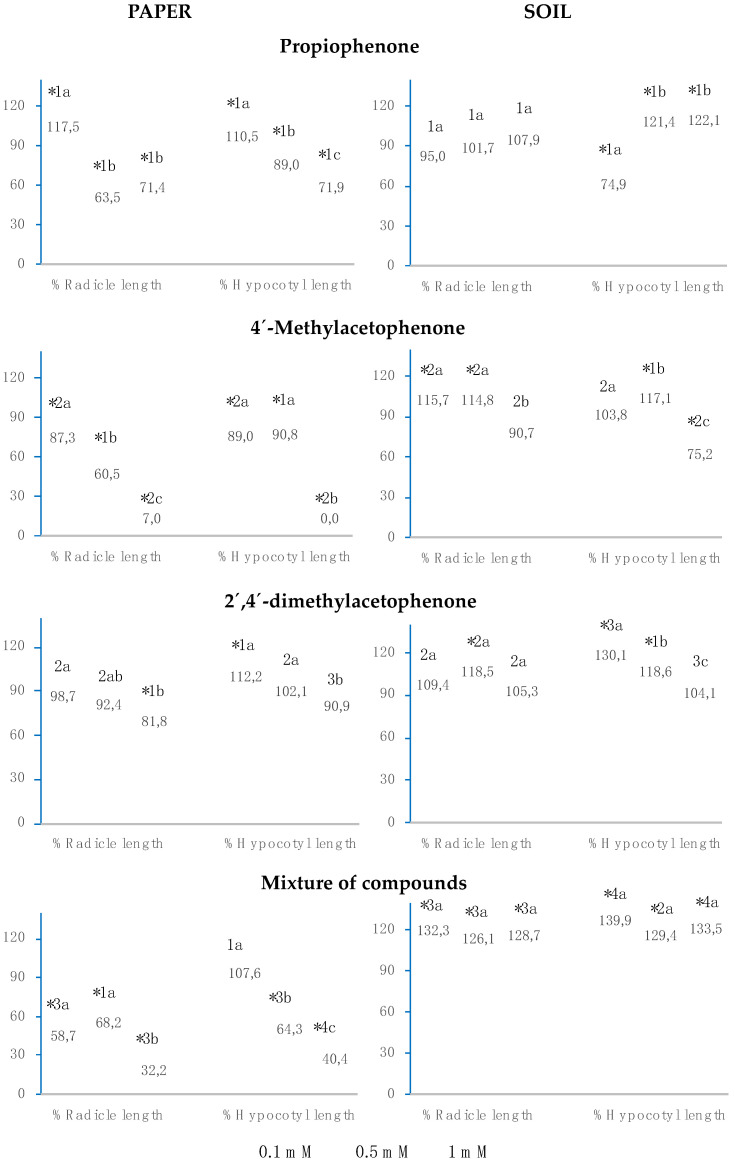
The effect of different concentrations of propiophenone, 4′-methylacetophenone, 2′,4′-dimethylacetophenone, and the mixture of the three compounds on the radicle and hypocotyl length of *Allium cepa*, expressed as the percentage relative to the control. Four replicates of each treatment were performed (n = 4 × 50 = 200 seeds in total for each solution). * Significantly different from the controls; 1, 2, 3, and 4: differences numbers indicate significant differences between treatments of the same index and for each concentration. The a, b, c: differences in small letters indicate significant differences between concentrations of the same index and for each treatment. *p* < 0.05 (Mann–Whitney U test).

## Data Availability

The data presented in this study are available on request from the corresponding author.

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
