# Peer review of "Evaluation of Propiophenone, 4-Methylacetophenone and 2′,4′-Dimethylacetophenone as Phytotoxic Compounds of Labdanum Oil from Cistus ladanifer L."

_plants, 2023, doi:10.3390/plants12051187_

Round 1

Reviewer 1 Report

The manuscript entitled “Evaluation of propiophenone, 4-methylacetophenone and 2,4-dimethylacetophenone as phytotoxic compounds of labdanum oil from Cistus ladanifer L.” covers one of the major concerns in agriculture today, the control of weeds that are resistant to current synthetic herbicides, and in a more sustainable way. To this end, they carried out a well-conducted preliminary in vitro assay where they tested the phytotoxicity of three compounds in isolation and the mixture of them. Very appropriately, they introduced soil, as it is a very important factor to take into account in Allelopathy studies. The fact that the mixture of the compounds and the soil increased the phytotoxicity is very interesting regarding further applications as an herbicide.

However, and taking into account that there are not many papers in the literature that compare paper vs. soil, it strikes me that other recent works with a very very similar methodology and results have not been cited at all. Pardo-Muras et al. have studied the phytotoxic effect of different phenolic and volatile compounds in isolation and their mixture, on paper and in soil, also obtaining higher effects with the mixture and in soil. I highly recommend reading and citing the following papers:

-Cytisus scoparius and Ulex europaeus Produce Volatile Organic Compounds with Powerful Synergistic Herbicidal Effects. Molecules 2019, 24, 4539; doi:10.3390/molecules24244539

- Complex Synergistic Interactions among Volatile and Phenolic Compounds Underlie the Effectiveness of Allelopathic Residues Added to the Soil for Weed Control. Plants 202211(9), 1114; https://doi.org/10.3390/plants11091114

Other specific comments:

Introduction:

-          Line 61: “…can be the solution…” may sound a bit pretentious, considering that this work, as I said before, is a preliminary in vitro study with not very noticeable inhibitory effects on a target species sensitive to allelochemicals. Of course it is an important first study for searching new herbicides, but there is still a lot of work to be done before making this claim.

-          Line 65: add the authority of the species at the first mention in the text. Apply in other cases through the manuscript.

-          Line 76 to 80: is confusing

Discussion:

the greater effect observed with the mixture of compounds and on the soil would need to be further discussed. After all, I consider that these are the main results of this work.

Methodology:

It is not made clear at any point whether the data represent the mean of the values, but I assume that they do. However, the growth parameters should be calculated as the median of the values, not the mean. This is because those seedlings, where radicle and coleoptile were measured, do not start from pregerminated seeds with the same radicle length, but from seeds where the time of germination was affected by the effect of the compounds. Applying the median corrects this error.

Author Response

The Referee is right and we corrected the revised manuscript accordingly.
Regarding the methodology, growth parameters were calculated as the mean length of radicle or hypocotyls of 40 seedlings (10 seedlings from each Petri dish for 4 replicates). The mean is calculated to follow the instructions of the citations referenced in the manuscript. If we calculate the median, the data does not vary significantly. Beyond the average value shown in the graph, what is important is the result of the statistical analysis. To carry out the statistical analysis, 40 measurements from the treatment plates (10 measurements per replicate) are confronted with another 40 from the control plates. Therefore, the significant differences will not change even if we change the mean value to the median. For this reason, it has been decided to maintain the average value, but if the referee deems it convenient and provides us with references, we can change the average to the median.

The authors thank the referee for his contributions.

Reviewer 2 Report

I find the study interesting, it deals with an important topic and brings valuable scientific results. The manuscript itself is well structured and written. However, my biggest reservations concern the use of only one model organism, Lactuca sativa - in this case, the results should not be presented as a general phytotoxic effect of the agents used, but rather a phytotoxic effect against only one specific model organism. I would suggest to accept MS for publication after additional experiments with at least one other model organism, f.e from the group of monocotyledonous plants, which can lead to obtaining of different results. But, in this way the results would become more widely applicable and significance of content would undobtedly increase.

Author Response

The Referee is right and we have included experiments with another monocot plant model organism (Allium cepa). These data were not entered earlier because they seemed contradictory. The effects of the compounds on onion were different, but with them the study is more complete. The authors thank the referee for his contributions.

Reviewer 3 Report

I have received the Manuscript (Manuscript Number: plants-2203124) entitled: ‘Evaluation of propiophenone, 4-methylacetophenone and 2,4´-dimethylacetophenone as phytotoxic compounds of labdanum oil from Cistus ladanifer L.’ submitted to Plants for a review.

The authors of the Manuscript described a research focused on evaluation of the phytotoxic activity of three phenolic compounds present in the essential oil of Cistus ladanifer toward lettuce seeds. The scientific hypothesis is defined and the methodology regarding experimental part used in this Manuscript are generally clear. However, manuscript current form contains a lot errors/misleading fragments and requires substantial improvement before publication:

· Line 21: delete “efficient”

· Lines 24-29: This is a poorly developed fragment with repeating sentences, which decreases quality of Manuscript. it is difficult to follow the conclusions in abstract in current form. I recommend to discuss the paper on the first place entirely, then describe separate compound as well as mixture. Authors should also provide the name of the plant that was selected for study.

· Line 48: replace “synthesis” for  “synthesize”

· Line 59-61: appropriate citation is  necessary after this sentence

· Line 75: replace “is constituted by” for “contain”

· Line 76-79: this sentence is written in a confusing way. I suggest to rephrase it and divide in to separate, but more simple sentences

· Line 94: the context of the verb “suffer” is misleading

· Figure 2 and 3: the value bars should be clearly visible and the current form is difficult to read. Moreover, OY axis is not defined in both.

· Line 166: “germination and seedling growth of” - germination was discussed in 2,1 section

· Line 167-170: “root and shoot”, “radicle and hypocotyl”, “radicle and cotyledon” - authors are using different names for the same parameter which is confusing and misleading, please choose one and be consistent within the whole document

· Line 216: replace ‘natural spaces’ for ‘natural areas’

· Line 216-221: this part requires appropriate citation

· Line 231: “derivatives retain the reaction site” - this construction is unclear

· Line 267: more reactive with what? this one is also confusing

· Line 310-312: this sentence requires improvement

· Line 374: please provide correct name of supplier along with the location of headquarters

· Please check the units within the whole document, because they are inconsistent, e.g. some are with a space before the value and some are not; instead of ‘ml’ there should be ‘mL’, etc.

· Line 393: “… were calculated.” – how? please provide appropriate details and equations

· Line 394: “radicle and shoot was measured” - how it was measured? can you provide more details?

· Line 401: instead of ‘(IC50)’ there should be ‘(IC50)’

Author Response

The Referee is right and we corrected the revised manuscript accordingly. The authors thank the referee for his contributions.

Round 2

Reviewer 2 Report

I highly appreciate the inclusion of another model organism and thus the additional work done by the authors, and I recommend paper to be accepted in present form.

Reviewer 3 Report

The authors have implemented a lot of improvements to the document. The corrected version of the Manuscript is suitable for publication in Plants